# DP-SGD-LF: Improving Utility under Differentially Private Learning via Layer Freezing

## Abstract

Differentially Private SGD (DP-SGD) is a widely known substitute for SGD to train deep learning models with privacy guarantees. However, privacy guarantees come at cost in model utility. The key DP-SGD steps responsible for this utility cost are per-sample gradient clipping, which introduces bias, and adding noise to the aggregated (clipped) gradients, which increases the variance of model updates. Inspired by the observation that different layers in a neural network often converge at different rates following a bottom-up pattern, we incorporate layer freezing into DP-SGD to increase model utility at fixed privacy budget. Through theoretical analysis and empirical evidence we show that layer freezing improves model utility, by reducing both the bias and variance introduced by gradient clipping and noising. These improvements in turn lead to better model accuracy, and empirically generalize over multiple datasets, models, and privacy budgets.

## 1 Introduction

Deep Neural Networks (DNNs) have seen a growing success at many tasks under various domains in recent years. As a result DNNs are now deployed in numerous applications, including some involving sensitive data, such as users' medical history, purchasing records, or chat histories. In these sensitive applications, data privacy is a concern. However, there is strong evidence that deep learning models memorize, and thus leak, information about their training data (Shokri et al., 2016; Carlini et al., 2020; 2022; Feldman & Zhang, 2020). To prevent data leakage, common DNN training algorithms such as Stochastic Gradient Descent (SGD) and its variants have been adapted to enforce Differential Privacy (DP) Song et al. (2013); Dwork et al. (2006), a rigorous privacy guarantee which provably mitigates data leaks. As a convenient drop-in replacement for SGD, the DP-SGD algorithm is commonly used for privacy-preserving machine learning, and numerous efforts haave improved its theoretical privacy analysis (Abadi et al., 2016; Mironov, 2017; Mironov et al., 2019).

However, the privacy guarantees offered by DP-SGD still come at a substantial cost in model utility (accuracy), despite substantial practical improvements over time De et al. (2022); Papernot et al. (2021). There are two key changes to SGD that DP-SGD introduces in each model update step. Each change is required to prove privacy guarantees, and contributes to utility costs. The first change is to clip per-sample gradient to a fixed $L_2$ norm bound, which introduces bias in the estimation of the gradient descent direction. The second change is to add noise from a standard Gaussian to the aggregated (clipped) gradients, which increases the variance of model updates.

We show using theoretical analysis that increasing the gradient clipping norm of a given DNN layer in DP-SGD reduces the variance introduced by DP noise and, under some assumptions, the clipping bias as well. Both lead to better convergence upper-bounds for DP-SGD. We combine this result with the observation that different layers in a DNN trained with SGD converge at different rates following a bottom-up pattern—which we empirically verify also holds for DP-SGD—and introduce the DP-SGD Layer Freeze (DP-SGD-LF) algorithm. This algorithm freezes the lower layers (closer to the input) of a DNN towards the end of training, which increases the norm of clipped gradients for the remaining layers, thereby decreasing the bias and variance introduced by DP-SGD when updating these parameters. Since the remaining layers benefit more from updates at this point of traininig, the finial accuracy increases.

We apply DP-SGD-LF to state of the art DP-SGD implementations on three datasets De et al. (2022); Papernot et al. (2021), and show that it improves the final model's accuracy by up to $1.3$ percentage

points, and is particularly effective in the high privacy (low DP $\epsilon$) regime. We also show that DP-SGD-LF is not sensitive to hyper-parameters, and propose and use easy to set, reasonable defaults.

The rest of the paper describes our contributions: after introducing the necessary background in §2, §3 introduces our algorithm, and supports its design through empirical and theoretical analysis. §4 then empirically confirms the expected behavior, and shows that DP-SGD-LF improves the accuracy of different models over multiple image classification datasets.

## 2 BACKGROUND

Mini-batch SGD is one of the most commonly used optimization algorithm in non-private deep learning. For each iteration $t$, and calling $\eta_t$ is the step size, SGD updates the parameters of the model $\theta$ by stepping into the direction of steepest descent, estimated with the averaged gradients over $B$ samples in a mini-batch,

$$\theta_{t+1} \leftarrow \theta_t - \eta_t \left[ \frac{1}{B} \sum_{i=1}^{B} g_t(x_i) \right]. \tag{1}$$

The convergence analysis of the SGD algorithm often rely on the following fundamental result.

**Lemma 2.1** (Decent Lemma (Bottou et al., 2018))**.** *Assuming the objective function $f : \mathbb{R}^d \to \mathbb{R}$ to be continuously differentiable and the gradient of $f$, $\nabla f : \mathbb{R}^d \to \mathbb{R}^d$ to be Lipschitz continuous with the Lipschitz constant $L > 0$, $\|\nabla f(v) - \nabla f(w)\| \leq L\|v - w\|$, $\forall v, w$, then $f(v) \leq f(w) + \nabla f(w)^T (v - w) + \frac{L}{2}\|v - w\|^2$, $\forall v, w$.*

Under privacy constraints, the DP-SGD algorithm provides a convenient substitution to SGD for training DNNs with differential privacy guarantees (Abadi et al., 2016). The DP-SGD algorithm protects privacy by clipping the per-sample gradient vector, $g_t(x_i) \leftarrow \nabla_{\theta_t} f(\theta_t, x_i)$, and adding noise drawn from a Normal distribution to the aggregated clipped gradients. Let $C$ be the L2-norm clipping threshold, $\sigma$ be the noise multiplier, and $d$ be the dimension of the model's parameters. the update rule for DP-SGD in each iteration is:

$$\theta_{t+1} \leftarrow \theta_t - \eta_t \left[ \frac{1}{B} \left( \sum_{i=1}^{B} \text{clip}\big(g_t(x_i), C\big) + z_t \right) \right], \ z_t \sim \mathcal{N}\big(0, \sigma^2 C^2 \mathbb{I}^d\big)$$

$$\text{clip}\big(g_t(x_i), C\big) \leftarrow g_t(x_i) / \max\left(1, \frac{\|g_t(x_i)\|_2}{C}\right), \tag{2}$$

where $C$ controls the maximum influence that an individual sample can have on the gradient (the sensitivity), and $\sigma$ controls the noise level scaled with respect to the sensitivity. We use the analysis based on Rényi Differential Privacy (RDP) (Mironov, 2017) for privacy accounting. The composition over $t$ steps of training and the conversion of the RDP guarantee to the $(\epsilon, \delta)$-DP guarantee follow from the results in Mironov et al. (2019). we use the publicly available implementation of the RDP privacy accountant in Opacus (Yousefpour et al., 2021).

## 3 DIFFERENTIALLY PRIVATE LEARNING WITH LAYER FREEZING

We propose to incorporate layer freezing with DP-SGD, and demonstrate its effectiveness in increasing the trained model's predictive accuracy at fixed privacy budget. The intuition behind the performance gain is as follows. The two key steps in DP-SGD, clipping and noising, provide a DP guarantee at the cost of degrading model utility: clipping potentially introduces bias into the estimated descent direction, since it truncates individual gradients before aggregation to control sensitivity (Chen et al., 2021; Pichapati et al., 2019; Zhang et al., 2019); noising introduces variance on top of the biased estimate by adding random noise to the aggregated clipped gradients. Freezing parameters limits the model capacity in learning representations, but could bring benefits by reducing the bias and variance caused by clipping and noising on the remaining trainable parameters. Given the observation that lower layers (closer to the input side) converge faster than higher layers (closer to the prediction), we can freeze the parameters in lower layers during training, to minimally sacrifice model capacity in exchange for the benefits of better updates for the upper layer parameters.

We detail our approach in the rest of this section. §3.1 shows the algorithm we propose. §3.2 presents empirical evidence that lower layers in a DP-SGD trained neural network converge faster than the upper layers. §3.3 shows theoretically that clipping and noising can be expected to degrade the convergence of model training, and presents our key metric to quantify this negative impact: the distortion angle in estimating the descent direction. In §3.4, we show that under some assumptions, layer freezing can improve both bias and variance caused by clipping and noising with respect to the trainable parameters. §4, then empirically evaluates our claims and demonstrates the effectiveness of layer freezing in improving model utility under multiple settings.

## 3.1 THE DP-SGD-LF ALGORITHM

Algorithm 1 shows the pseudo code of our method. When the current iteration index $t$ exceeds a preset threshold $t^f$, we apply parameter freezing on the first $m^f$ layers (closest to the input) of the model architecture. $t^f$ and $m^f$ are two hyperparameters of the algorithm. In §4, we empirically show that the performance is not sensitive to these hyperparameters in a wide range of values, and provide generic defaults that we use in experiments. Before freezing, the algorithm behaves exactly as DP-SGD. After freezing ($t > t^f$), the frozen layers' gradients are set to 0 before clipping, and are not noised. The frozen parameters are not updated. The rest of the parameters are updated as usual, following Equation 2. Since layer freezing is decided by hyper-parameters without adapting to the data, the privacy analysis of DP-SGD-LF is identical to that of DP-SGD, and existing privacy accounting can be re-used.

---

**Algorithm 1:** DP-SGD-LF

**Output:** Model parameters $\theta$
**Input:** Dataset $D = (x_i, y_i)_{i=0}^N$, loss function $f$, learning rate $\eta_t$, batch size $B$, noise multiplier $\sigma$,
       L2-norm clipping threshold $C$, privacy parameters $\epsilon, \delta$, freezing parameters $t^f, m^f$

Initialize $\theta_0$ randomly
Calculate the total number of iterations $T(\epsilon, \delta, B, N, \sigma)$
**for** $t = 1 \ldots, T$ **do**
    Take a random batch with sampling probability $B/N$
    **if** $t > t^f$ **then**
        Partition $\theta_t$ into $\theta_t^{\text{frozen}}$ and $\theta_t^{\text{trainable}}$ according to the first $m^f$ layers
        Update $\theta_{t+1}^{\text{trainable}}$ as in Equation 2
        Merge $\theta_t^{\text{frozen}}$ with $\theta_{t+1}^{\text{trainable}}$ as $\theta_{t+1}$
    **else**
        Update $\theta_{t+1}$ as in Equation 2
    **end**
**end**

---

## 3.2 LAYER CONVERGENCE FOLLOWS A BOTTOM-UP PATTERN IN PRIVATE TRAINING

There is strong empirical evidence to suggest that for DNNs trained in the non-private setting, layers in the DNN architecture converge at different rates, and exhibit a bottom-up pattern (Wang et al., 2022; Raghu et al., 2017; Morcos et al., 2018; Yosinski et al., 2014; Rogers et al., 2020). In this section, we verify that a similar observation holds under private training. The dataset, model, and algorithm used for demonstration are CIFAR-10, a 5-layer CNN (with the last layer being a softmax-activated classification layer) and DP-SGD with $C = 3$ and $\sigma = 1$. The accuracy of the full model is around 0.63 at the last training step, when $\epsilon$ is around 7.

We examine the privacy-utility trade-off of the DP-SGD trained model by training only a single layer after $\epsilon = 3$ where the model achieves moderately high accuracy but is not fully trained yet. We note that the final layer is the softmax activated classification layer and is not frozen in these experiments. As shown in Figure 1 (Left), we observe that there is a minor gain in accuracy when only Layer 1 is trained further in steps while a larger increase is observed when only upper layers are trained. The final accuracy obtained by training Layer 1 only is also considerably lower than when only training the other layers. To support this observation, we also measure the convergence quality using a post-hoc analysis tool, PWCCA (Morcos et al., 2018), which compares different layers'

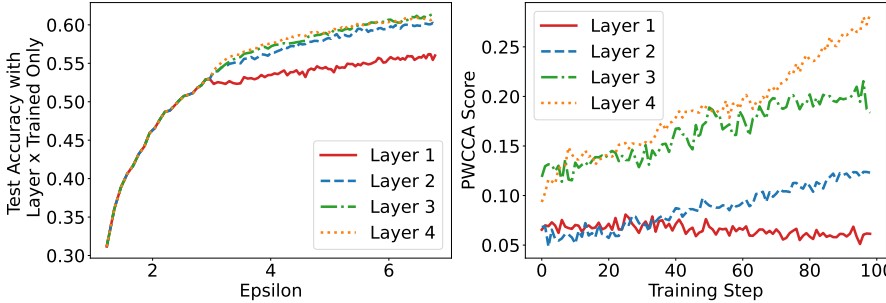

Figure 1: **Left:** The privacy-utility trade-off when only a single layer is trained with every other layer frozen (except for the final classification layer) after $\epsilon = 3$. Layer 1 has the least amount of gain in accuracy when other layers are frozen while the upper layers perform similarly. **Right:** The PWCCA score over training steps for different layers. A lower PWCCA score indicates the model converges better. Layer 1 shows a weak sign of convergence while the upper layers has no strong evidence of converging.

intermediate activation vectors throughout training to the converged activation vector from a fully-trained model (i.e., the final iteration the model converges to, in which the accuracy is reasonably high). Figure 1 (Right) shows the PWCCA score for the first 4 layers over training steps. A lower PWCCA score means that the layer converges better. We observe that only Layer 1 shows a weak sign of convergence, while the upper layers are likely to be dominated by the accumulated noise and have no clear sign of convergence. These experiments confirm that: lower layers converge better and earlier during training with DP-SGD, and focusing training on higher layers leads to more utility.

### 3.3 DISTORTIONS IN DESCENT DIRECTION DEGRADES DP-SGD PERFORMANCE

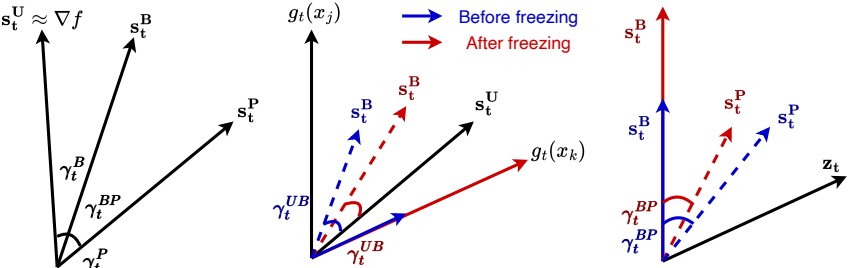

Figure 2: **Left:** An example of the true ($\nabla f$), unbiased ($s_t^U$), biased ($s_t^B$), private ($s_t^P$) signals and the distortion angles ($\gamma_t^{UP}, \gamma_t^{UB}, \gamma_t^{BP}$) as in Definition 3.1 and 3.2. **Middle:** Assuming the unclipped and clipped gradients remain unclipped and clipped after freezing, when $s_t^U$ is more aligned with the clipped gradients, then $\gamma_t^{UB}$ is reduced after freezing since the clipped gradient is clipped less (magnitude increase prior to freezing) due to an decrease in $\|g_t(x_j)\|$. **Right:** For an arbitrary noise vector, since $s_t^B$ increases in magnitude $\gamma_t^{BP}$ is reduced after freezing such that the noising makes $s_t^P$ less variable in direction.

We first define the true, unbiased, biased and private signal vectors (which refer to the true descent direction, and different estimates of it from a minibatch: without changes, with clipping, and with clipping and noise, respectively) and the corresponding distortion angle in each step $t$ of the optimization algorithm. Figure 2 (Left) shows a demonstration using 2-dimensional vectors. We adapt the Decent Lemma (Lemma 2.1) to DP-SGD, and show that bias and variance in the distortion angle will increase the upper-bound on convergence. This suggests that a larger bias (mis-oriented descent direction) and higher variance in the distortion angle is likely to lead to slower convergence.

**Definition 3.1** (The True, Unbiased, Biased and Private Signal Vectors). *For each step $t$, the **true signal vector** $\nabla f_t$ is the gradient of the loss function $f$ evaluated at $\theta_t$ on all the training data. It*

*is the true direction of steepest descent for the empirical loss. The **unbiased signal vector** $s_t^U$ is the mean gradient over samples in a batch drawn following a uniform sampling scheme. It is an unbiased estimate of $\nabla f_t$ (Bottou et al., 2018), $s_t^U := \frac{1}{B} \sum_{i=0}^{B} g_t(x_i)$. The **biased signal vector** $s_t^B$ is the mean of sample-wise clipped gradients on a batch drawn following a uniform sampling scheme. It is a biased estimate of $s_t^U$ since per-sample gradients are re-scaled differently before aggregation, $s_t^B := \frac{1}{B} \sum_{i=0}^{B} \text{clip}(g_t(x_i), C)$. Let $z_t \sim \mathcal{N}(0, (1/B^2)\sigma^2 C^2 \mathbb{I})$ be the random noise vector, the **private signal vector** is the sum of $s_t^B$ and $z_t$ and is the actual descent direction in DP-SGD updates, $s_t^P := s_t^B + z_t$.*

With Definition 3.1, the parameter update rule of DP-SGD can be rewritten as:

$$\theta_{t+1} \leftarrow \theta_t - \eta_t s_t^P, \quad s_t^P = s_t^B + z_t. \tag{3}$$

We note that gradient clipping in DP-SGD has been shown to be biased and a bias vector $b_t$ can be decomposed following the analysis of Chen et al. (2021). We define the distortion in estimating the descent direction by the angle between $\nabla f_t$ and $s_t^P$.

**Definition 3.2** (Distortion Angle). *For each step $t$, the **distortion angle** $\gamma_t$ is the angle between $\nabla f_t$ and $s_t^P$,*

$$\gamma_t := \arccos\left( \frac{\langle \nabla f_t, s_t^P \rangle}{\|\nabla f_t\| \|s_t^P\|} \right),$$

*where $\langle , \rangle$ represents the dot product and $\|\cdot\|$ represents the L2-norm. $\gamma_t$ can be decomposed into $\gamma_t^B$ and $\gamma_t^{BP}$ which is the angle between $\nabla f_t$ and $s_t^B$, and between $s_t^B$ and $s_t^P$, respectively: $\gamma_t = \gamma_t^B + \gamma_t^{BP}$.*

In what follows, the expectations and variances of $s_t^U$, $s_t^B$, $s_t^P$, $\gamma_t$, $\gamma_t^B$, and $\gamma_t^{BP}$ are taken with respect to the data sampling and the DP noise distribution when applicable.

We show next show that the bias and variance introduced by clipping and noising leads to a worse convergence bound rate bound for DP-SGD. The proof of Lemma 3.1 is in Appendix A.

**Lemma 3.1** (DP-SGD convergence bound). *Following the proof of Bottou et al. (2018) for DP-SGD, we show the following result. Assuming the objective function $f : \mathbb{R}^d \to \mathbb{R}$ to be continuously differentiable and the private gradient of $f$, $\nabla f : \mathbb{R}^d \to \mathbb{R}^d$ to be Lipschitz continuous with the Lipschitz constant $L > 0$, $\|\nabla f(v) - \nabla f(w)\| \leq L\|v - w\|, \forall v, w$, the convergence bound of DP-SGD is,*

$$\min_{t=0,\ldots,T} \mathbb{E}\big[\|\nabla f(\theta_t)\|^2\big] \leq \left( f(\theta_0) - \mathbb{E}\big[f(\theta_T)\big] \right.$$
$$\left. - \sum_{t=1}^{T} \eta_t \mathbb{E}\big[\cos(\gamma_t^B)\|\nabla f(\theta_t)\|\|s_t^B\|\big] + \sum_{t=1}^{T} \eta_t^2 \frac{L}{2B^2}\sigma_{DP}^2 C^2 \right) / \sum_{t=1}^{T} \eta_t.$$

Finally, we explain why $\gamma_t$ is an effective metric for utility. From the convergence bound above, we see the negative effect of $\gamma_t$ due clipping and noising: a larger bias ($\gamma_t^B$) and variance ($\sigma_{DP}$) make the upper-bound larger, and could potentially lead to worse model performance:

(1) A negative value of $\mathbb{E}\big[\cos(\gamma_t^B)\|\nabla f(\theta_t)\|\|s_t^B\|\big]$ could be caused by a large bias angle $\gamma_t^B$. This makes the convergence bound worse by adding a positive term. A smaller $\gamma_t^B$ ($|\gamma_t^B| \in [0, \frac{\pi}{2}]$) means that the DP-SGD descent direction is better aligned with the true direction of steepest descent ($\gamma_t$ is smaller), and $\mathbb{E}\big[\cos(\gamma_t^B)\|\nabla f(\theta_t)\|\|s_t^B\|\big]$ is positive. However, $|\gamma_t^B| > 0$ still makes $cos(\gamma_t^B) < 1$, increasing the bound.

(2) Since $z_t$ is drawn from a zero mean Normal distribution, adding noise does not bias the estimation. However, from the convergence bound we see that a larger variance of the noise $(L/2B^2)\sigma_{DP}^2 C^2$ also makes the bound worse. A higher variance means a larger noise is more likely added to $s_t^B$, therefore we would expect $\gamma_t$ to be large.

### 3.4 LAYER FREEZING MITIGATES DISTORTIONS IN OPTIMIZATION DIRECTION

In this section, we introduce (strong) assumptions, but supported by empirical measurements in Appendix E, under which we prove that freezing some parameters benefits the remaining trainable parameters, by reducing the bias and variance in their distortion angle $\gamma_t$. Intuitively, since a subset of the parameters are frozen, the gradient's $L2$-norm is reduced (the frozen parameters have a gradient of zero). This in turn leads to an increase in the magnitude of each sample-wise clipped gradients on the remaining parameters. Under our assumptions, clipping gradients less aggressively reduces bias in the distortion angle $\gamma_t^B$, since each clipped gradient is closer to its original value. A larger biased signal is also more robust to noise, and $\gamma_t^{BP}$ is smaller for any fixed noise draw, leading to reduced variance. Figure 2 (Middle and Right) demonstrates this intuition with 2-dimensional vectors, and the formal results are stated below, following from results which we prove in Appendix B and C.

**Lemma 3.2** (The magnitude of the biased signal of trainable parameters increases after freezing)**.** *Let $s_t^{B,b}(\theta'_t)$ and $s_t^{B,a}(\theta'_t)$ be the biased signal of the trainable parameters before and after freezing, then $\|g_t^a(x_i)\| \le \|g_t^b(x_i)\| \,\forall x_i$ and $\|s_t^{B,a}(\theta'_t)\| \ge \|s_t^{B,b}(\theta'_t)\|$.*

**Assumption 3.1.** *The mean of per-sample gradients $g(x_i)$ with a larger magnitude such that $\|g(x_i)\| > C$ are more aligned (smaller angle) with the gradient direction $\nabla f_t$, whereas those with a smaller magnitude such that $\|g(x_i)\| \le C$ are less aligned (larger angle).*

**Assumption 3.2.** *Rescaling the gradient norm of $\|g(x_i)\|$ by freezing does not change the gradients that are clipped and unclipped gradients, but only rescales the clipped gradients.*

**Proposition 3.1** (Freezing reduces bias and variance in distortion angle with respect to trainable parameters)**.** *Let $\theta_t$ be the set of full parameters and let $\theta'_t$ be the subset of trainable parameters. Let $\gamma_t(\theta'_t)$ be the distortion angle with respect to trainable parameters. Let the superscript $b$ and $a$ denotes the quantity **before** and **after** freezing occurs. Under Assumptions 3.1 and 3.2, the following results hold:*

$$(1)\ \mathbb{E}[\gamma_t^{B,a}(\theta'_t)] \le \mathbb{E}[\gamma_t^{B,b}(\theta'_t)];$$
$$(2)\ \mathrm{Var}[\gamma_t^{BP,a}(\theta'_t))] \le \mathrm{Var}[\gamma_t^{BP,b}(\theta'_t))];$$
$$(3)\ \mathbb{E}[\gamma_t^a(\theta'_t)] \le \mathbb{E}[\gamma_t^b(\theta'_t)],\ \mathrm{Var}[\gamma_t^a(\theta'_t)] \le \mathrm{Var}[\gamma_t^b(\theta'_t)]$$

*Proof.* (1) By the clipping function in Equation 2, each per-sample gradient is either clipped if $\|g_t(x_i)\| > C$ or preserved to its original value if $\|g_t(x_i)\| \le C$. Let $g'$ denote the gradients with respect to trainable parameters, $v$ and $w$ be the vector of the sum of unclipped and clipped gradients in a random batch of size $B$ respectively,

$$v = \sum_{j:\|g'_t(x_j)\| \le C} \left( g'_t(x_j) \right),\ w = \sum_{k:\|g'_t(x_k)\| > C} \left( \frac{g'_t(x_k) \cdot C}{\|g'_t(x_k)\|} \right),\ |j| + |k| = B.$$

By Lemma 3.2, since $\|g_t'^a(x_k)\| \le \|g_t'^b(x_k)\| \,\forall i$, $(g'_t(x_k) \cdot C)/\|g_t'^a(x_k)\|^a \ge (g'_t(x_k) \cdot C)/\|g_t'^b(x_k)\|$, i.e. the magnitude of the clipped gradients increase after freezing comparing to before. By Assumption 3.2, $v$ does not change. Since $w^a \ge w^b$ we have:

$$\mathbb{E}\left[ \arccos\left( \frac{\langle v, v + w^a \rangle}{\|v\|\|v + w^a\|} \right) \right] \le \mathbb{E}\left[ \arccos\left( \frac{\langle v, v + w^b \rangle}{\|v\|\|v + w^b\|} \right) \right],$$

which means $\mathbb{E}(\gamma_t^{vB,a}) \le \mathbb{E}(\gamma_t^{vB,b})$. Under Assumption 3.1, if $\mathbb{E}(\gamma_t^{Uv,a}) \ge \mathbb{E}(\gamma_t^{vw,b})$, then we have $\mathbb{E}[\gamma_t^{B,a}(\theta'_t)] \le \mathbb{E}[\gamma_t^{B,b}(\theta'_t)]$.

(2) Since the noise is drawn from a zero mean Normal distribution, for each batch, when taking expectation over the random noise draw, $\mathbb{E}[\gamma_t^{BP}] = 0$. The variance term can be simplified as,

$$\mathrm{Var}[\gamma_t^{BP}] = \mathbb{E}[(\gamma_t^{BP})^2] - (\mathbb{E}[\gamma_t^{BP}])^2 = \mathbb{E}\left[ \arccos\left( \frac{\langle s_t^B(\theta'_t), s_t^B(\theta'_t) + z_t \rangle}{\|s_t^B(\theta'_t)\|\|s_t^B(\theta'_t) + z_t\|} \right)^2 \right].$$

Let $X$ denote the random variable inside $\arccos(\cdot)$, for every batch the distribution of the private gradient is the same as the random noise distribution,

$$\mathrm{Var}[\gamma_t^{BP}] = \int_{-\infty}^{\infty} (\arccos(x))^2 f(x)dx, \ X \sim \mathcal{N}(0, \sigma^2 C^2 \mathbb{I}^d).$$

We show the following result in Appendix C,

$$\mathbb{E}\left[ \arccos\left( \frac{\langle s_t^B(\theta_t')^a, s_t^B(\theta_t')^a + z_t \rangle}{\|s_t^B(\theta_t')^a\|\|s_t^B(\theta_t')^a + z_t\|} \right)^2 \right] \leq \mathbb{E}\left[ \arccos\left( \frac{\langle s_t^B(\theta_t')^b, s_t^B(\theta_t')^b + z_t \rangle}{\|s_t^B(\theta_t')^b\|\|s_t^B(\theta_t')^b + z_t\|} \right)^2 \right].$$

Therefore $\mathrm{Var}[\gamma_t^{BP,a}(\theta_t')] \leq \mathrm{Var}[\gamma_t^{BP,b}(\theta_t')]$.

(3) is a consequence of (1) and (2), since $\mathbb{E}[\gamma_t^a(\theta_t')] = \mathbb{E}[\gamma_t^{B,a}(\theta_t')] + \mathbb{E}[\gamma_t^{BP}(\theta_t')] = \mathbb{E}[\gamma_t^{B,a}(\theta_t')]$ since $\mathbb{E}[\gamma_t^{BP}(\theta_t')] = 0$. $\mathrm{Var}[\gamma_t(\theta_t')] = \mathrm{Var}[\gamma_t^B(\theta_t')] + \mathrm{Var}[\gamma_t^{BP}(\theta_t')]$ since the noise $z_t$ is drawn independently. By Assumption 3.1, $\forall$ batch with size B, $\gamma_t^{B,a} \leq \gamma_t^{B,b}$, so $\mathrm{Var}[\gamma_t^{B,a}(\theta_t')] \leq \mathrm{Var}[\gamma_t^{B,b}(\theta_t')]$ thus $\mathrm{Var}[\gamma_t^a(\theta_t')] \leq \mathrm{Var}[\gamma_t^b(\theta_t')]$. $\square$

# 4 EVALUATING LAYER FREEZING IN END-TO-END PRIVATE TRAINING

In this section we empirically examine our method on utility improvements over multiple datasets, models and privacy levels. We also empirically evaluate the claims from previous sections. We also perform a sensitivity analysis on hyperparameters, and present our suggested defaults. Unless otherwise specified, the analysis results in this section are demonstrated using CIFAR-10 and a 5-layered CNN model.

| Dataset | Model | Test Acc. | | | | | $\delta$ |
|---------|-------|-----------|--------------|--------------|--------------|--------------|-----------|
| | | LF (ours) | $\epsilon = 1$ | $\epsilon = 2$ | $\epsilon = 3$ | $\epsilon = 7$ | |
| MNIST | CNN | ✗ | 0.956 (0.002) | 0.975 (0.002) | 0.981 ($< 0.001$) | / | $10^{-5}$ |
| | | ✓ | **0.957 (0.001)** | **0.976 (0.002)** | 0.981 ($< 0.001$) | / | |
| FashionMNIST | CNN | ✗ | 0.801 (0.005) | 0.855 (0.007) | 0.862 (0.002) | / | $10^{-5}$ |
| | | ✓ | **0.815 (0.005)** | **0.868 (0.002)** | **0.875 (0.002)** | / | |
| CIFAR10 | CNN | ✗ | / | / | 0.533 (0.010) | 0.637 (0.005) | $10^{-5}$ |
| | | ✓ | / | / | **0.548 (0.003)** | **0.651 (0.003)** | |
| CIFAR10 | Wide-ResNet | ✗ | 0.558 (0.252) | 0.625 | / | / | $10^{-5}$ |
| | | ✓ | **0.571 (0.120)** | **0.636** | / | / | |

Table 1: A summary of the results training with DP-SGD with and without layer freezing. The results are the median and standard deviation over 5 independent runs. The bold results indicate better performance.

**Baseline models and experimental setup.** We implement layer freezing on top of the existing baseline models. For MNIST, FashionMNIST and CIFAR10 with CNN, we use the baseline model from Papernot et al. (2021). For the CIFAR10 experiment with Wide-ResNet, the baseline model is from De et al. (2022). All hyperparameters related to the model, training, or DP settings are kept constant between with and without layer freezing (and tuned without freezing in the baselines' papers). For each experiment, we run the model with and without layer freezing 5 times independently, using different random seeds. For the CIFAR10 model with Wide-ResNet we were only able to repeat the experiments for $\epsilon = 1$ settings due to limited computational resources. The exact hyperparameters and other details are included in Appendix D.

**Model utility under DP-SGD-LF.** Table 1 compares the performances of DP-SGD-LF on top of the current 'state-of-the-art' baselines and show the median and standard deviation over 5 runs. We observe that layer freezing generally improves the predictive performance across datasets, models and privacy budgets, as it improves the median score and has a smaller standard deviation.

**Layer Freezing Improves Angle of Distortions in Optimization Direction** Figure 3 shows the median of $|s_t^B|$ of the trainable layers after freezing the first 3 layers after step 5000. In the last step, the model accuracy is around 0.64, for $\epsilon = 7$. We observe an increase in signal strength for both

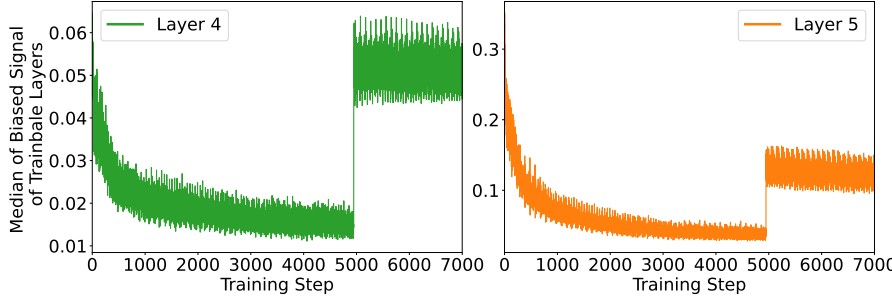

Figure 3: The median of $\|s_t^B\|$ for trainable layers over training steps after the lower 3 layers frozen at step 5000. We observe in both Layer 4 and 5 there is an increase in the signal strength.

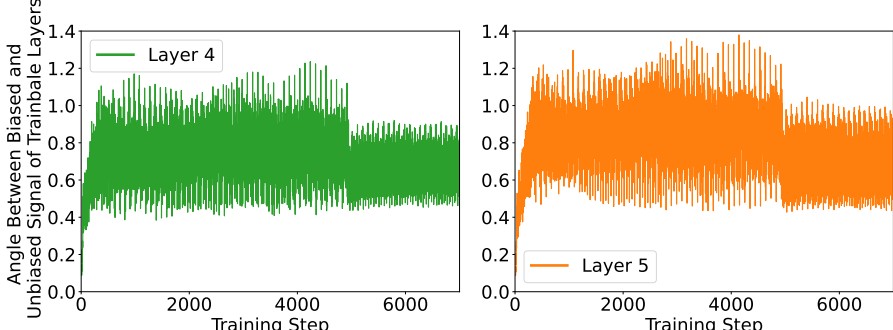

Figure 4: The change in $\gamma_t^{UB}$ for trainable layers after freezing the lower 3 layers at step 5000. We observe a decrease in $\gamma_t^{UB}$ in both Layer 4 and 5, indicating that the biased signal is closer to the unbiased signal after freezing.

layers which matches the claim in §3.4. Figures 4 and 5 show the change in the distortion angle when layer freezing is imposed. We note that since the full-sample-size gradient is expensive to compute, we measure the angle between the unclipped and unnoised SGD gradients $\gamma_t^{UB}$ on the same batch of data. We thus make the underlying assumption that moving the DP descent direction closer to the minibatch direction in each iteration $t$ would improve the private training performance. We observe that both $\gamma_t^{UB}$ and $\gamma_t^{BP}$ decrease after freezing for trainable layers. The distortion measured in $\gamma_t^{UB}$ is generally weaker than in $\gamma_t^{BP}$ as the absolute scale is higher in the later. Although the privacy hyperparamters $C$ and $\sigma$ affect the results, we generally observe that noising has a stronger negative effect than clipping in terms of distorting the optimization direction.

**The tradeoff between gains of freezing and model capacity.** The benefits of freezing layers comes at a cost in model capacity, which potentially limits performance. We evaluate such a tradeoff

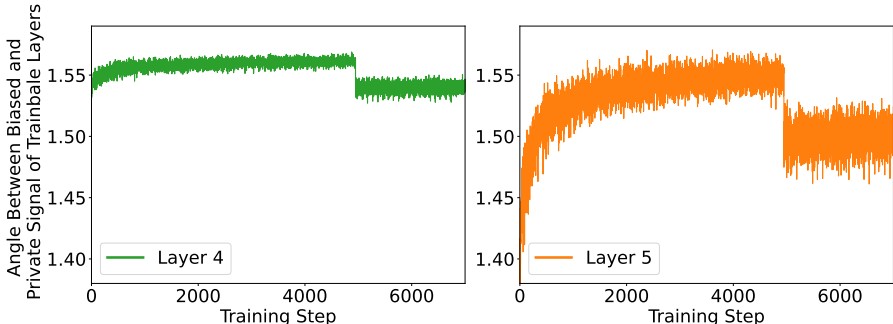

Figure 5: The change in $\gamma_t^{BP}$ for trainable layers after freezing the lower 3 layers at step 5000. We observe a decrease in $\gamma_t^{UB}$ in both Layer 4 and 5, indicating that the improved strength in signal makes it more robust to noise as noising changes the direction less.

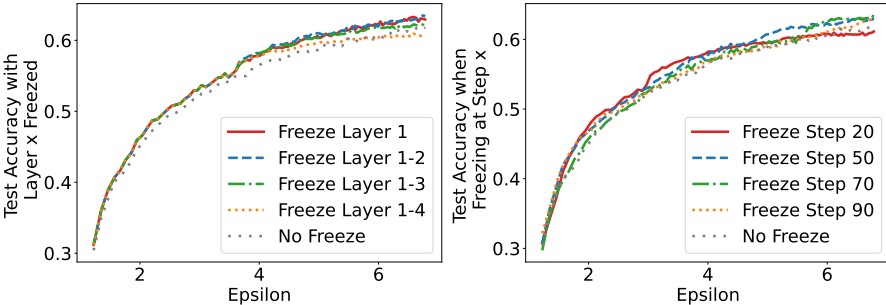

Figure 6: Evaluating the hyperparameter choices of how many layers to freeze ($m^f$) and when to start freezing ($t^f$). **Left:**The privacy-utility trade-off when freezing different number of layers after $\epsilon = 3$. **Right:**The privacy-utility trade-off when freezing the lower 3 layers at different steps.

by measuring the final model accuracy under different choices of freezing hyperparameters: how many layers to freeze ($n^f$) and when to start freezing ($m^f$). Figure 6 (Left) compares the model's utility when freezing different number of layers. In these experiments, freezing starts after $\epsilon = 3$. We observe that in general, the choice of how many layers to freeze is not quite sensitive as long as the model still has a reasonable capacity (e.g., leaving only the last layer is insufficient). Figure 6 (Right) compares the model utility by varying when to start freezing layers. In these experiments, we freeze Layers 1-3 at different training steps. We observe that freezing early can result in worse performance, whereas freezing in later steps does not bring any gains prior to freezing.

Determining the optimal values for $t^f$ and $m^f$ might lead to additional privacy cost. Since the utility is not too sensitive to these hyperparameters in a wide reasonable range, we suggest the following default hyperparameters, which are those we use for all experiments in Table 1: given a target privacy budget $\epsilon$, we calculate the number of training to take (based DP and optimization parameters), and set $t^f$ to be about 20 steps earlier. We set $m^f$ to be the lower half of all layers in a model.

## 5 RELATED WORK

There is a rich literature on representations learning in DNNs under non-private training. It is commonly observed that the lower level layers extract more general features and are easier to train while higher level layers capture more abstract and task-specific features and take more steps for learning good representations (Raghu et al., 2017; Morcos et al., 2018; Yosinski et al., 2014; Rogers et al., 2020). Wang et al. (2022) demonstrate with image classification tasks that different layers converge with a bottom-up pattern. This line of work promotes the possibility of transfer learning, with a subset of the parameters to be inherited from pre-trained models, and kept frozen when fine-tuning on downstream tasks. In non-private training, parameter freezing is mainly used to reduce data requirements, computational costs, or communication (Zhuang et al., 2019).

Parameter has also been studied in the private training setting, also under the transfer learning scenario but with additional privacy benefits. A subset of the model parameters are transferred from publicly trained models, and are frozen when fine-tuning with privat data on the downstream task. Such an approach is empirically effective across multiple computer vision and natural language processing tasks (Tramèr & Boneh, 2021; Luo et al., 2021; Yu et al., 2021; Li et al., 2021; Mehta et al., 2022). In these work, the frozen parameters are either used directly as good initializations, or attached to additional layers for finetuning. In Luo et al. (2021), the parameters chosen to be frozen are the ones with smaller scales which are usually considered less important in a neural network. It coincides with our observation that freezing the lower level layers, which converge early, does not overly hurt model performance.

A few work closely related works incorporate techniques to increase sparsity in private learning. Zhang et al. (2021) presents a theoretical study on the benefits of sparse gradients in wide neural network models trained with DP. Talwar et al. (2015) studies the DP LASSO model which encourages sparsity by design. Huang et al. (2020) shows that pruning can be an alternative approach to privacy.

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

## A  PROOF OF LEMMA 3.1

**Lemma** (DP-SGD convergence bound). *Following the proof of Bottou et al. (2018) for DP-SGD, we show the following result. Assuming the objective function $f : \mathbb{R}^d \to \mathbb{R}$ to be continuously differentiable and the private gradient of $f$, $\nabla f : \mathbb{R}^d \to \mathbb{R}^d$ to be Lipschitz continuous with the Lipschitz constant $L > 0$, $\|\nabla f(v) - \nabla f(w)\| \le L\|v - w\|$, $\forall v, w$, the convergence bound of DP-SGD is,*

$$\min_{t=0,\ldots,T} \mathbb{E}\big[\|\nabla f(\theta_t)\|^2\big] \le \left( f(\theta_0) - \mathbb{E}[f(\theta_T)] - \sum_{t=1}^{T} \eta_t \mathbb{E}\big[\nabla f(\theta_t)^T b_t\big] + \sum_{t=1}^{T} \eta_t^2 \frac{L}{2B^2} \sigma^2 C^2 \right) / \sum_{t=1}^{T} \eta_t.$$

*Proof.* Let $f$ be the loss function we want to optimize, let $\nabla f(\theta_t)$ be the true steepest descent gradient vector, and let $s_t^U$, $s_t^P$ be the unbiased and private gradient as in Definition 3.1 at step $t$. Since the private gradients are assumed to be $L$-Lipschitz continuous, the descent lemma (§2) implies that,

$$f(\theta_{t+1}) \le f(\theta_t) + \nabla f(\theta_t)^T (\theta_{t+1} - \theta_t) + \frac{L}{2} \|\theta_{t+1} - \theta_t\|^2.$$

Substituting in the parameter update rule of DP-SGD, $\theta_{t+1} \leftarrow \theta_t - \eta_t s_t^P$, we get,

$$f(\theta_{t+1}) \le f(\theta_t) - \eta_t \nabla f(\theta_t)^T s_t^P + \eta_t^2 \frac{L}{2} \|s_t^P\|^2.$$

Taking the expectation over the data distribution and assuming the step size $\eta_t$ is independent of which data are sampled in each iteration $t$ we get,

$$\mathbb{E}\big[f(\theta_{t+1})\big] \le f(\theta_t) - \eta_t \nabla f(\theta_t)^T \mathbb{E}\big[s_t^P\big] + \eta_t^2 \frac{L}{2} \mathbb{E}\big[\|s_t^P\|^2\big].$$

The private gradient $s_t^P$ is composed of the biased gradient $s_t^B$ and the random noise $z_t$. The bias in $s_t^B$ can be isolated and quantified by integrating over the probability density function of the gradient noise caused by data sampling (Chen et al., 2021). Therefore we can simplify the expectation of $s_t^P$ as,

$$\mathbb{E}\big[s_t^P\big] = \mathbb{E}\big[s_t^U\big] + b_t + \mathbb{E}\big[z_t\big] = \nabla f(\theta_t) + b_t,$$

for some bias vector $b_t$. The second equality holds because $s_t^U$ is an unbiased estimator of the true gradient $\nabla f(\theta_t)$ with the assumption that each $x_i$ is drawn with a uniform sampling scheme as in the regular mini-batch SGD, and $\mathbb{E}[z_t] = 0$ since $z_t \sim \mathcal{N}\big(0, (1/B^2)\sigma^2 C^2 \mathbb{I}\big)$. The variance of $s_t^P$ is bounded under DP-SGD since we performed gradient clipping to control sensitivity and added the noise drawn from Normal distribution. Given batch size $B$, L2-clipping norm threshold $C$ and noise multiplier $\sigma_{DP}$,

$$\mathbb{E}\big[\|s_t^P\|^2\big] \le \frac{1}{B^2} \sigma_{DP}^2 C^2.$$

Therefore, the progress bound for each parameter update step $t$ of DP-SGD is,

$$\mathbb{E}\big[f(\theta_{t+1})\big] \le f(\theta_t) - \eta_t \big(\|\nabla f(\theta_t)\|^2 + \nabla f(\theta_t)^T b_t\big) + \eta_t^2 \frac{L}{2B^2} \sigma_{DP}^2 C^2.$$

Rearranging the terms and summing over all iterations $t = 1, \ldots, T$ we get,

$$\sum_{t=1}^{T} \left( \eta_t \|\nabla f(\theta_t)\|^2 \right) \le \sum_{t=1}^{T} \left( f(\theta_t) - \mathbb{E}\big[f(\theta_{t+1})\big] - \eta_t \nabla f(\theta_t)^T b_t + \eta_t^2 \frac{L}{2B^2} \sigma_{DP}^2 C^2 \right).$$

Taking expectations on both sides and simplify using the Law of Iterated Expectations we get,

$$\sum_{t=1}^{T} \eta_t \mathbb{E}\big[\|\nabla f(\theta_t)\|^2\big] \leq \sum_{t=1}^{T} \Big( \mathbb{E}\big[f(\theta_t)\big] - \mathbb{E}\big[f(\theta_{t+1})\big] \Big) - \sum_{t=1}^{T} \eta_t \mathbb{E}\big[\nabla f(\theta_t)^T b_t\big] + \sum_{t=1}^{T} \eta_t^2 \frac{L}{2B^2} \sigma_{DP}^2 C^2.$$

Similar to the usual gradient descent scenario, when the magnitude of the gradient vector shrinks and gets closer to 0, we consider the algorithm is converged (to optima or to saddle point). We can rewrite the above inequality in terms of the smallest gradient norm over all training steps as,

$$\min_{t=0,\ldots,T} \mathbb{E}\big[\|\nabla f(\theta_t)\|^2\big] \sum_{t=1}^{T} \eta_t$$

$$\leq \sum_{t=1}^{T} \eta_t \mathbb{E}\big[\|\nabla f(\theta_t)\|^2\big]$$

$$\leq \sum_{t=1}^{T} \Big( \mathbb{E}\big[f(\theta_t)\big] - \mathbb{E}\big[f(\theta_{t+1})\big] \Big) - \sum_{t=1}^{T} \eta_t \mathbb{E}\big[\nabla f(\theta_t)^T b_t\big] + \sum_{t=1}^{T} \eta_t^2 \frac{L}{2B^2} \sigma_{DP}^2 C^2$$

$$\leq f(\theta_0) - \mathbb{E}\big[f(\theta_T)\big] - \sum_{t=1}^{T} \eta_t \mathbb{E}\big[\nabla f(\theta_t)^T b_t\big] + \sum_{t=1}^{T} \eta_t^2 \frac{L}{2B^2} \sigma_{DP}^2 C^2.$$

Let $s_t^B$ and $\gamma_t^B$ be defined as in Definition 3.1 and 3.2, we can rewrite the bound in terms of these quantities,

$$\mathbb{E}[\nabla f(\theta_t)^T b_t] = \mathbb{E}[\langle \nabla f(\theta_t), s_t^B \rangle] = \mathbb{E}\big[\cos(\gamma_t^B)\|\nabla f(\theta_t)\|\|s_t^B\|\big],$$

Therefore the convergence bound of DP-SGD is,

$$\min_{t=0,\ldots,T} \mathbb{E}\big[\|\nabla f(\theta_t)\|^2\big] \leq \bigg( f(\theta_0) - \mathbb{E}\big[f(\theta_T)\big] - \sum_{t=1}^{T} \eta_t \mathbb{E}\big[\cos(\gamma_t^B)\|\nabla f(\theta_t)\|\|s_t^B\|\big]$$

$$+ \sum_{t=1}^{T} \eta_t^2 \frac{L}{2B^2} \sigma_{DP}^2 C^2 \bigg) / \sum_{t=1}^{T} \eta_t.$$

$\square$

## B    PROOF OF LEMMA 3.2

**Lemma** (The magnitude of the biased signal of trainable parameters increases after freezing). *Let $s_t^{B,b}(\theta_t')$ and $s_t^{B,a}(\theta_t')$ be the biased signal of the trainable parameters before and after freezing, then $\|g_t^a(x_i)\| \leq \|g_t^b(x_i)\| \; \forall x_i$ and $\|s_t^{B,a}(\theta_t')\| \geq \|s_t^{B,b}(\theta_t')\|$.*

*Proof.* Let $g_t(x_i, \theta_t)$, $g_t(x_i, \overline{\theta_t'})$ and $g_t(x_i, \theta_t')$ be the corresponding gradient vector of the full, frozen and trainable parameters for each instance $x_i$, since $\theta_t = \overline{\theta_t'} \cup \theta_t'$ then

$$g_t^b(x_i, \theta_t) = g_t^b(x_i, \overline{\theta_t'}) \cup g_t^b(x_i, \theta_t'), \; g_t^a(x_i, \theta_t) = g_t^a(x_i, \theta_t') = g_t^b(x_i, \theta_t').$$

By the Triangle Inequality, we get,

$$\|g_t^b(x_i, \theta_t)\| = \|g_t(x_i, \overline{\theta_t'}) + g_t^b(x_i, \theta_t')\| \leq \|g_t^b(x_i, \overline{\theta_t'})\| + \|g_t^b(x_i, \theta_t')\|.$$

Since $\|g_t^b(x_i, \overline{\theta_t'})\| \geq 0$, $\|g_t^b(x_i, \theta_t)\| \geq \|g_t^b(x_i, \theta_t')\| = \|g_t^a(x_i, \theta_t)\|$. By Definition 3.1,

$$s_t^{B,b}(\theta_t') = \frac{1}{B} \sum_{i=1}^{B} g_t(x_i) / \max\left(1, \frac{\|g^b(x_i, \theta_t')\|}{C}\right),$$

$$s_t^{B,a}(\theta_t') = \frac{1}{B} \sum_{i=1}^{B} g_t(x_i) / \max\left(1, \frac{\|g^a(x_i, \theta_t')\|}{C}\right),$$

so it follows that $\|g_t^a(x_i)\| \leq \|g_t^b(x_i)\| \; \forall x_i$ and $\|s_t^{B,a}(\theta_t')\| \geq \|s_t^{B,a}(\theta_t')\|$.  $\square$

## C    SUPPORTING PROOFS FOR PROPOSITION 3.1

We first show a common result that helps the proofs stated afterwards: The cosine similarity increases between two vectors if either one of the vectors is a sum of the other vector plus a third vector with a larger magnitude. Let $v_1$ and $v_2$ be two arbitrary vectors, let $v_3$ be the sum of $v_1$ and $v_2$, $v_3 = v_1 + v_2$. Keeping $v_1$ the same, if the magnitude of $v_2$ increases, then the cosine similarity between $v_1$ and $v_3$ is larger, leading to a smaller angle between $v_1$ and $v_3$. The same result also holds if $v_2$ is kept the same and the magnitude of $v_1$ increases.

Note that if $v_1$, $v_2$ and $v_3$ are all 2-dimensional vectors, then the claims follow naturally from the Parallelogram law. We show that the claim holds in the general $p$-dimensional case. Let $\mathbf{v_i} = [v_{i1}, v_{i2}, \ldots, v_{ip}]$ be the flattened $p$-dimensional vector, we show one direction of the results which when $v_1$ is kept the same and the magnitude of $v_2$ increases, $v_2' = kv_2$ for some $k \geq 1$, and the other direction follows by symmetry. we show that

$$LHS = \frac{\langle \mathbf{v_2}, (\mathbf{v_2} + \mathbf{v_1}) \rangle}{\|\mathbf{v_2}\| \|(\mathbf{v_2} + \mathbf{v_1})\|} \geq \frac{\langle \mathbf{v_2'}, (\mathbf{v_2'} + \mathbf{v_1}) \rangle}{\|\mathbf{v_2'}\| \|(\mathbf{v_2'} + \mathbf{v_1})\|} = RHS.$$

Expanding the dot product and the norm into summations on both sides, and let

$$a = \sum_i v_{2i}^2, \ b = \sum_i v_{1i} v_{2i}, \ c = \sum_i v_{1i}^2.$$

Then we can simplify LHS and RHS as,

$$LHS = \frac{k^2 a + kb}{k\sqrt{a} + \sqrt{k^2 a + c + 2kb}}, \ RHS = \frac{a + b}{\sqrt{a} + \sqrt{a + c + 2b}}.$$

Since $a \geq 0$, $c \geq 0$, if $b \geq 0$, we can easily verify that,

$$(1) \text{At } k = 1, LHS - RHS = 0; \ (2) \ \lim_{k \to \infty} (LHS - RHS) = \infty; \ (3) \ \nabla_k (LHS - RHS) \geq 0.$$

Therefore, $LHS - RHS$ is a positive non-decreasing function in $k$ for $k \geq 1$, i.e. $LHS \geq RHS$.

Finally, we show that the following result is true,

$$\mathbb{E}\left[ \arccos\left( \frac{\langle s_t^B(\theta_t')^a, s_t^B(\theta_t')^a + z_t \rangle}{\|s_t^B(\theta_t')^a\| \|s_t^B(\theta_t')^a + z_t\|} \right)^2 \right] \leq \mathbb{E}\left[ \arccos\left( \frac{\langle s_t^B(\theta_t')^b, s_t^B(\theta_t')^b + z_t \rangle}{\|s_t^B(\theta_t')^b\| \|s_t^B(\theta_t')^b + z_t\|} \right)^2 \right].$$

Let X denote the random variable inside the $\arccos$ function, in this case X follows the same distribution as the noise, $X \sim \mathcal{N}(0, \sigma^2 C^2 \mathbb{I}^d)$ since for each fixed $s_t^B$ the sources of randomness comes from the addition of random noise. Let $x$ be a sample of X and let $x^b$ and $x^a$ denote the function calculating the quantity of x before and after freezing,

$$x^a = \frac{\langle s_t^B(\theta_t')^a, s_t^B(\theta_t')^a + z_t \rangle}{\|s_t^B(\theta_t')^a\| \|s_t^B(\theta_t')^a + z_t\|}, \ x^b = \frac{\langle s_t^B(\theta_t')^b, s_t^B(\theta_t')^b + z_t \rangle}{\|s_t^B(\theta_t')^b\| \|s_t^B(\theta_t')^b + z_t\|}.$$

Since $x^a \geq x^b$, $\arccos(x^a)^2 \leq \arccos(x^b)^2$, therefore,

$$\int_{-\infty}^{\infty} \arccos(x^a)^2 f(x) dx \leq \int_{-\infty}^{-\infty} \arccos(x^b)^2 f(x) dx,$$

where $f(x) = (1/(\sigma'\sqrt{2\pi})) \exp(-(1/2)((x-\mu)/\sigma')^2)$, $\mu = 0$, $\sigma'^2 = \sigma^2 C^2 \mathbb{I}^d$ is the probability density function of X. Therefore $\mathbb{E}[(\gamma_t^{BP})^2]^a \leq \mathbb{E}[(\gamma_t^{BP})^2]^b$.

## D    EXPERIMENT DETAILS

We implement our method and the CNN baseline model (Papernot et al., 2021) with JAX (Bradbury et al., 2018). The MNIST model is run using a 2-layered CNN model, FashionMNIST and CIFAR-10 are run using a 5-layered CNN model. The architecture is the same as in Table 1 and 2 in Papernot et al. (2021). The Wide-ResNet baseline model is from Balle et al. (2022) and we implement the layer freezing part on top of it. We mostly adopt the hyperparameters suggested in the original paper. Below are the details for each experiment:

(1) MNIST, 2-layered CNN: $C = 1.0$, $\sigma_{DP} = 1.923$, $B = 2048$, $\eta = 2.0$, activation function is tempered sigmoid with scale=1.58, inverse temp=3.0, offset=0.71 as suggested in the original paper. When using layer freezing, we freeze the first 2 layers after epoch 3, 17, 39 for $\epsilon = 1, 2, 3$ respectively.

(2) FashionMNIST, 5-layered CNN: $C = 1.0$, $\sigma_{DP} = 2.15$, $B = 2048$, $\eta = 4.0$, activation function is tempered sigmoid with scale=1.58, inverse temp=3.0, offset=0.71 as suggested in the original paper. When using layer freezing, we freeze the first 3 layers after epoch 4, 19, 40 for $\epsilon = 1, 2, 3$ respectively.

(3) CIFAR10, 5-layered CNN: For the $\epsilon = 3$ experiment we used $C = 3.0$, $\sigma_{DP} = 1.0$, $B = 512$, $\eta = 0.15$, activation function is tempered sigmoid with scale=1.58, inverse temp=3.0, offset=0.71 as suggested in the original paper. We freeze the first 3 layers after epoch 20. For the $\epsilon = 7$ experiment we used $C = 1.0$, $\sigma_{DP} = 1.47$, $B = 2048$, $\eta = 4.0$, We freeze the first 3 layers after epoch 75.

(4) CIFAR10, Wide-Resnet: We adopt the publicly available configuration file in Balle et al. (2022). We freeze the first 2 convolution groups after update step 800 and 3000 for $\epsilon = 1, 2$ respectively.

## E  EMPIRICALLY EVALUATING ASSUMPTION 3.1 AND 3.2

In this section we empirically examine the assumptions in Section 3.4. We observe that Assumption 3.1 is generally well-supported empirically as the sum of clipped gradients are closer, in terms of having a smaller angle, to the true gradient direction $\nabla f_t$ than the sum of unclipped gradients. We also observe that Assumption 3.2 is generally not true since freezing rescales $\|g(x_i)\|$ thus moving some per-sample gradient $g(x_i)$ from being clipped to unclipped. However, since only a small amount of $g(x_i)$ are moved, the directions of the summed clipped and unclipped gradients after freezing are quite aligned with the directions before freezing.

Intuitively, if we clip less by freezing a subset of the parameters, under Assumption 3.1, we assign a larger weight to the clipped gradients which are more aligned with the truth so that we decrease the bias. To verify Assumption 3.1, in each iteration we record the clipping status and compute the sum of the clipped (those with $\|g(x_i)\| \geq C$) and unclipped ($\|g(x_i)\| < C$) gradients, and compute the distortion angle (as in Definition 3.2) to the true gradient $\nabla f_t$ computed on all training examples. Figure 7 shows the results with different L2-clipping norm $C$ run with DP-SGD, CIFAR10, $\sigma_{DP} = 1.0$, batch size $B = 512$ and learning rate $\eta = 0.15$. At the end of training, $\epsilon = 7.0$ and the test accuracy is $0.42, 0.58, 0.60, 0.58$ for $C = 0.1, 1, 5, 10$ respectively. We observe that in general the summed clipped gradients are more aligned with the true gradient than the summed unclipped gradients by having a smaller angle for different $C$s which empirically supports Assumption 3.1. It matches with our intuition since there are often more gradients being clipped than unclipped throughout training, given a reasonable range of $C$ from 0.1 to 10 as suggest by the Opacus authors (Yousefpour et al., 2021), and these per-sample gradients with larger magnitudes are more likely to dominate the true gradient direction.

Assumption 3.2 is used to characterize sample gradients that change clipping status due to layer freezing, from clipped to unclipped. Figure 8 shows the change in the number of clipped and unclipped gradients. We see that a small number of gradients does change their category as some data points' gradients go from being clipped to unclipped when rescaling $\|g(x_i)\|$. Referring to the demo in Figure 2, Assumption 3.2 ensures that the directions of $g_t(x_j)$ and $g_t(x_k)$ (the summed unclipped and clipped gradients respectively) do not change after freezing so that the observed change $\gamma_{UB}$ is mostly due to rescaling the gradient norm which the analysis proceeds. We empirically measure the change in angle before and after freezing, for the sum of gradients in the clipped and unclipped category. We observe that the change in the angle of the summed clipped gradients are 0.09 and 0.11 radians, and 0.10 and 0.12 radians for the summed unclipped gradients, when freezing the first 3 layers of parameters after epoch 20 and 40. This matches our intuition that the direction of the sum of gradients over a minibatch is stable to changing a small number of medium-sized gradients from the clipped to the unclipped category.

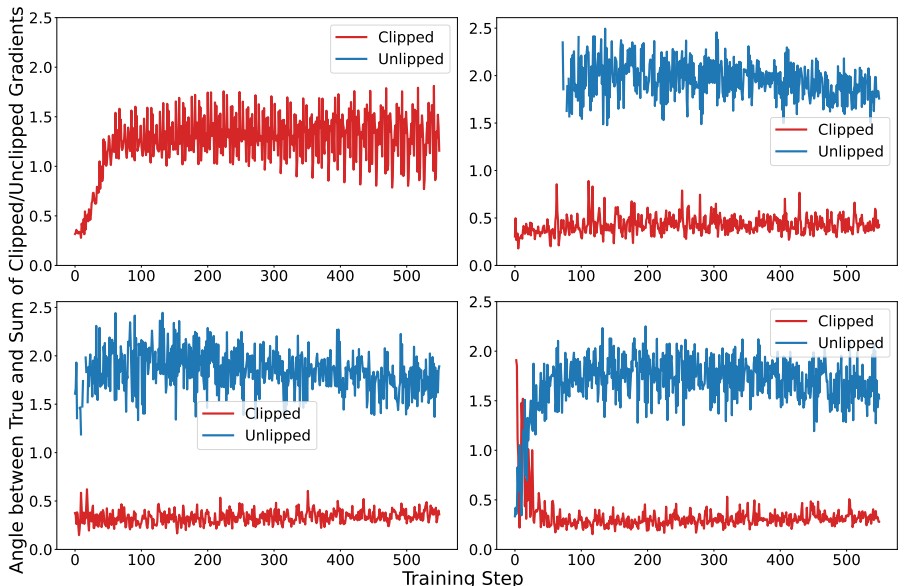

Figure 7: Comparing the angle between the true gradient and the sum of clipped and unclipped gradients for different $C$. **Top-Left:** $C = 0.1$, **Top-Right:** $C = 1.0$, **Bottom-Left:** $C = 5.0$, **Bottom-Right:** $C = 10.0$. Missing values indicate at step $t$ all the gradients are clipped. We observe that in general the clipped gradients are more aligned with the true gradient by having a smaller angle across $C$.

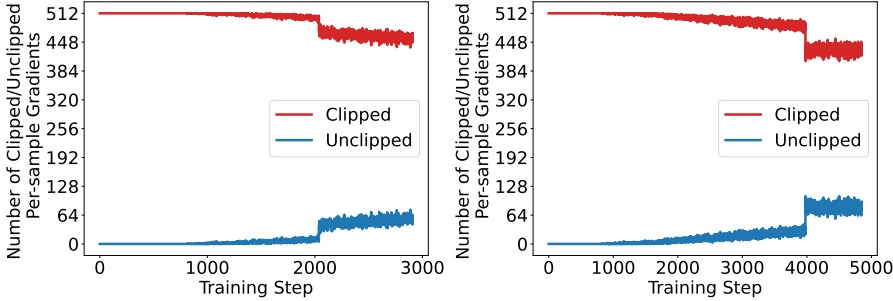

Figure 8: The number of clipped and unclipped gradients when freezing after **Left:** epoch 20 and **Right:** epoch 40. A small number of gradients changed status from being clipped to unclipped after freezing.

