# OpenReview forum: "DP-SGD-LF: Improving Utility under Differentially Private Learning via Layer Freezing"
_ICLR.cc/2023/Conference — Submitted to ICLR 2023_

### Official Review · Reviewer_z6fv · 2022-10-23

**Confidence:** 3
**Correctness:** 3
**Technical Novelty And Significance:** 2
**Empirical Novelty And Significance:** 2
**Recommendation:** 6

**Clarity, Quality, Novelty And Reproducibility:**

The paper is easy to follow and in good quality, the novelty is moderate significant and the experiment should be reproducible with details provided in the paper.

**Strength And Weaknesses:**

Strengths:
1. Freezing bottom layers during training is proven to be useful in DP training in a few other works, this paper provides some theory support of the approach.

2. The decomposition of bias caused by clipping from an angular perspective is new in the community, to the best of my knowledge.

Weaknesses:
1. It is unclear whether the bias caused by clipping in Lemma 3.1 is a tight characterization or not, if the authors could provide a case when the bias is achieved, it will make the theory more convincing.

2. The experiments are only for relatively simple architectures and datasets, if the authors could verify that freezing lower layers will benefit more variety of tasks, it would strengthen the contribution.

**Summary Of The Paper:**

The paper studies the effect of freezing parameters after certain iterations of differentially private training, both theoretically and empirically. Theory shows the convergence could be improved under certain situations and experiments show freezing layers closer to input (bottom layers) after training for a while could improve performance after DP training.



**Summary Of The Review:**

Overall I feel the paper studied an interesting problem/approach, with both theory and experiment support, but there is plenty room for improvements.

---

> ### Author Response · Authors · 2022-11-19
> **Response to Reviewer z6fv**
>
> Thank you for the constructive comments. We added Appendix E for more details about the assumptions in Section 3.4.
>
> *It is unclear whether the bias caused by clipping in Lemma 3.1 is a tight characterization or not, if the authors could provide a case when the bias is achieved, it will make the theory more convincing.*
>
> We would not claim that Lemma 3.1 finds the tightest upper bound for the convergence of DP-SGD. The purpose of Lemma 3.1 is mainly to show that (1) the clipping bias could lead to slower convergence, and (2) the distortion angle (as in Definition 3.2) could be used as a metric for evaluating clipping bias as they are positively related. There are many existing evidences on per-sample gradient clipping (DP-SGD without the noising part) hinders convergence. Two extreme cases could be seen from Example 1 and 2 in Section 1 of [1].
>
> *The experiments are only for relatively simple architectures and datasets, if the authors could verify that freezing lower layers will benefit more variety of tasks, it would strengthen the contribution.*
>
> We used the architectures and datasets following the current state-of-the-art papers [2, 3], we will certainly consider including more datasets and tasks in an extended work.
>
> [1] Xiangyi Chen, Zhiwei Steven Wu, and Mingyi Hong. 2020. Understanding gradient clipping in private SGD: a geometric perspective.
>
> [2] Soham De, Leonard Berrada, Jamie Hayes, Samuel L. Smith, Borja Balle. 2022. Unlocking High-Accuracy Differentially Private Image Classification through Scale.
>
> [3] Papernot, Nicolas \& Thakurta, Abhradeep \& Song, Shuang \& Chien, Steve \& Erlingsson, Úlfar. (2020). Tempered Sigmoid Activations for Deep Learning with Differential Privacy.

---

### Official Review · Reviewer_PEmL · 2022-10-25

**Confidence:** 5
**Correctness:** 2
**Technical Novelty And Significance:** 1
**Empirical Novelty And Significance:** 1
**Recommendation:** 3

**Clarity, Quality, Novelty And Reproducibility:**

The overall quality is okay, especially the clarity. However, I doubt the originality of the work is significant given that last-layer training (also known as linear probing) and training only the last few layers is commonly used already (see Sec 5.2 in https://aclanthology.org/2020.blackboxnlp-1.4.pdf and "top2" method in https://arxiv.org/pdf/2110.05679.pdf)

**Strength And Weaknesses:**

Strength: The paper is clearly written and the algorithm is carefully verified, from both theoretical and empirical perspectives.

Weakness: The major flaw is in the novelty which is explained in detail in the following section. Another major flaw is the scope of this work: notice that DP is a systematic effort, covering different clipping function (e.g. recent DP-NSGD or automatic clipping still work with layer freezing, but not mentioned; also per-layer clipping will likely violate Assumption 3.2; but this work only considers the common gradient clipping in the flat clipping style), fine-tuning methods, tasks (this work only experiment on tiny images and not consider language tasks), optimzers (only experiment with SGD, not adaptive optimizers which have been used to achieve SOTA on CIFAR10), etc. Therefore, this work really studies a very narrow perspective of DP deep learning.

Because of the narrow scope of this work, I would expect to see better analysis or empirical improvement. However, the improvement is insignificant (about 1% within the incomplete Table 1), and the analysis relies on too strong assumptions that are not empirically verified. In fact, it shouldn't be hard to print out the percentage of per-sample gradients that are clipped/not-clipped before and after freezing.

**Summary Of The Paper:**

This paper studies the DP-SGD with a set of fine-tuning methods, which freeze some hidden layers in a neural network. The authors claim that their method can (under some strong assumptions) improve the utility under the same privacy budget, which is verified on small image datasets.

**Summary Of The Review:**

I recommend heavy revision of this work before publishing.

---

> ### Author Response · Authors · 2022-11-19
> **Response to Reviewer PEmL**
>
> Thank you for the constructive comments. We added Appendix E for more details about the assumptions in Section 3.4.
>
> We would like to clarify our differences from freezing parameters in pre-training / fine-tuning cases. Freezing in pre-training / fine-tuning is very common in both non-private and private training, which the objectives are often to efficiently train downstream tasks with smaller models, warm-starting with prior information from pre-trained models, and to save privacy budget additionally under privacy-constrained scenarios. We study the benefits of freezing when training from scratch in DP, and show that even though the model is not as good as a non-DP model, there are benefits to freezing at the end of training. We showed that it might not be worth updating some parameters in private training at certain optimization stage since we pay a price of privately protecting their gradients (by introducing bias and variance in the optimization direction). The potential benefits of sparsity from freezing does not require a well-trained model, as we showed in Table 1 that a consistent increase in accuracy is observed for different $\epsilon$ values at different training steps. We consider it a more generic method for improving accuracy in end-to-end training, i.e. it might be worth freezing some parameters in the fine-tuning phase with frozen layers inherited from pre-trained models, or it might be worth freezing some parameters if releasing a privately trained pre-train model.
>
> We used the architectures and datasets following the current state-of-the-art papers [1, 2]. We did not include other optimizers because Adam is often underperformed by SGD especially in image classification tasks on datasets such as CIFAR10 in both non-private [3] and private [4] training scenarios. We will certainly consider including NLP tasks in an extended work.
>
> [1] Soham De, Leonard Berrada, Jamie Hayes, Samuel L. Smith, Borja Balle. 2022. Unlocking High-Accuracy Differentially Private Image Classification through Scale.
>
> [2] Papernot, Nicolas \& Thakurta, Abhradeep \& Song, Shuang \& Chien, Steve \& Erlingsson, Úlfar. (2020). Tempered Sigmoid Activations for Deep Learning with Differential Privacy.
>
> [3] Ashia C. Wilson, Rebecca Roelofs, Mitchell Stern, Nathan Srebro, Benjamin Recht. (2017). The Marginal Value of Adaptive Gradient Methods in Machine Learning.
>
> [4] Koskela, A. \&amp; Honkela, A.. (2020). Learning Rate Adaptation for Differentially Private Learning.

---

### Official Review · Reviewer_Hfmj · 2022-10-27

**Confidence:** 3
**Correctness:** 4
**Technical Novelty And Significance:** 2
**Empirical Novelty And Significance:** 2
**Recommendation:** 6

**Clarity, Quality, Novelty And Reproducibility:**

Clarity:
In general it's clear, but the theoretical analysis seems a bit confusing to me. (See point below.)

Quality:
- The analysis in Section 3.4 sounds quite straightforward to me. Because I think when we freeze some parameters, the norm "quota" of the rest of the parameters are for sure increased and that would give us reduced bias. Could you elaborate more on the difficulty of the analysis?
- The empirical results don't seem very significant.

Novelty:
I'm a bit concerned about this, as it seems like the idea of layer freezing is quite common in DPSGD training (though in the pre-training / fine-tuning case, as has been mentioned in the paper). Also, I feel like the works about per-layer clipping or adaptive clipping might be related, as they also look at properties of different layers. I guess per-layer clipping can be viewed as a general case of freezing, i.e. some layer has clip norm 0.

Reproducibility:
I didn't see the specification of batch size and number of steps in the experiments. Those parameters are quite essential as they affects the privacy-utility tradeoff.


Other comments:
"... adding noise does not bias the estimation...A higher variance means a larger noise is more likely added to s_t^B , therefore we would expect γ_t to be large.": It seems a bit contradictive to me. Also, I feel like if the noise is increased by increasing C, then γ_t^B might even be smaller, right?

- Assumption 3.2 "... does not change the gradients that are clipped and unclipped gradients": I don't understand the sentence. Could you elaborate more?

**Strength And Weaknesses:**

Strength:
The paper looks at an important problem, and provides some interesting empirical and theoretical analyses. I like the part about PWCCA specifically.

Weakness:
- Given the amount of work on layer freezing in private training (though in the pre-training / fine-tuning case, as has been mentioned in the paper), I'm not so sure about the novelty of the work.
- The empirical improvement doesn't seem very significant.

**Summary Of The Paper:**

The paper proposes to do layer freezing for DPSGD training. It analyzes the properties of different layers in the training process, provides some theoretical analysis on the affect of freezing, and shows the effectiveness of the algorithm in some real datasets.

**Summary Of The Review:**

Some analyses seem interesting, while I'm a bit concerned about the novelty of the work.

---

> ### Author Response · Authors · 2022-11-19
> **Response to Reviewer Hfmj**
>
> Thank you for the constructive comments. We add Appendix E for more details about the assumptions in Section 3.4. The hyperparameters used in experiments were included in Appendix D. We will release code for reproducibility purposes.
>
> *the norm "quota" of the rest of the parameters are for sure increased and that would give us reduced bias.*
>
> The first half is indeed straightforward, since the norm "quota" for the remaining parameters increases after freezing, but it does not directly lead to reduced bias in the angle. Intuitively this is because clipping bias does not have a clear relationship with how much the gradients are clipped, which can be seen from the experiments of [1] (Figure 5), the analysis of [2], and the fact that L2-norm clipping value $C$ often needs to be carefully tuned. We can consider an example violating Assumption 3.1 for intuition. Consider the case where the sum of unclipped gradients is more aligned with the true gradient than the sum of clipped gradients. This could possibly occur if $C$ is large and a few mislabeled examples have large, misaligned gradients. In this case, freezing would result in less clipping for those misaligned gradients, moving the minibatch gradient further from the true gradient, potentially increasing bias in the angle.
>
> *it seems like the idea of layer freezing is quite common in DPSGD training (though in the pre-training / fine-tuning case, as has been mentioned in the paper).*
>
> We would like to clarify our differences from freezing parameters in pre-training / fine-tuning cases. Freezing in pre-training / fine-tuning is very in both non-private and private training, with the objective to efficiently train downstream tasks with smaller models, warm-starting with prior information from pre-trained models, and to save privacy budget under privacy-constrained scenarios. We study the benefits of freezing when training from scratch in DP, and show that even though the model is not as good as a non-DP model, there are benefits to freezing at the end of training. We show that it might not be worth updating some parameters in private training at certain optimization stage since we pay a price of privately protecting their gradients (by introducing bias and variance in the optimization direction). The potential benefits of sparsity from freezing does not require a well-trained model, as we showed in Table 1 that a consistent increase in accuracy is observed for different $\epsilon$ values at different training steps. We consider it a more generic method for improving accuracy in end-to-end training, i.e. it might be worth freezing some parameters in the fine-tuning phase with frozen layers inherited from pre-trained models, or it might be worth freezing some parameters if releasing a privately trained pre-train model.
>
> *the works about per-layer clipping or adaptive clipping might be related, as they also look at properties of different layers. I guess per-layer clipping can be viewed as a general case of freezing, i.e. some layer has clip norm 0.*
>
> Adaptive clipping is related to freezing, and has a similar effect on gradient computation (shrinking the gradient norm allocated to a layer increases the norm quota available to other layers). The main difference between changing $C$ and freezing is that freezing has an impact on the noise as well, since we can avoid adding noise at all on frozen layers.
>
> *if the noise is increased by increasing C, then $\gamma_t^{B}$ might even be smaller*
>
> Clipping and noising distort the optimization direction in different ways: clipping 'changes' the optimization direction while noising makes the gradients more random. Since noise is drawn from $\mathbf{N}\big(0, \sigma^{2}C^{2}\mathbb{I}^{d}\big)$, the rotation symmetry means that the angle change is 0 in expectation (increasing the angle estimation variance, not bias). However, if we look at each realized sample draw, the probability of getting a specific noise value would follow the probability density function of $\mathbf{N}\big(0, \sigma^{2}C^{2}\mathbb{I}^{d}\big)$, which suggests a higher probability of sampling a large value if the variance term is larger. Therefore, adding noise drawn from a zero-mean normal distribution does not bias the estimation in expectation, but there is a higher chance to observe larger values in each sample draw. Since we decomposed the effect of bias and variance in the analysis, noising would have no effect on $\gamma_t^{B}$, and the effect of changing $C$ on $\gamma_t^{B}$ would follow as above, which there is no clear linear relationship between the size of the clipping bias and how much the gradients are clipped.
>
> [1] Martin Abadi, Andy Chu, Ian Goodfellow, H. Brendan McMahan, Ilya Mironov, Kunal Talwar, and Li Zhang. 2016. Deep Learning with Differential Privacy.
>
> [2] Xiangyi Chen, Zhiwei Steven Wu, and Mingyi Hong. 2020. Understanding gradient clipping in private SGD: a geometric perspective.

---

### Decision · Program_Chairs · 2023-01-20

**Decision:**

Reject

**Justification For Why Not Higher Score:**

Low novelty of investigating layer freezing, small effect size.

**Justification For Why Not Lower Score:**

N/A

**Metareview: Summary, Strengths And Weaknesses:**

This work investigates layer freezing during private training. Prefixes of the network are iteratively frozen as training proceeds. This leads to minor improvements in accuracy (i.e., absolute improvements on the order of 1%-2%). Some theoretical analysis/explanation is given.

Reviewers felt the method was a bit low in novelty (since freezing parameters is well studied), and the effect is quite small to stand alone as the main contribution. The theoretical analysis/perspective was of interest, but was not considered significant enough to warrant acceptance.